# Estimating the Total Number of Residential Fire-Related Incidents and Underreported Residential Fire Incidents in New South Wales, Australia by Using Linked Administrative Data

**DOI:** 10.3390/ijerph18136921

**Published:** 2021-06-28

**Authors:** Nargess Ghassempour, W. Kathy Tannous, Gulay Avsar, Kingsley E. Agho, Lara A. Harvey

**Affiliations:** 1School of Business, Western Sydney University, Parramatta, NSW 2150, Australia; K.tannous@westernsydney.edu.au (W.K.T.); G.avsar@westernsydney.edu.au (G.A.); 2Rozetta Institute, The Rocks, NSW 2000, Australia; 3Translational Health Research Institute, Western Sydney University, Campbelltown, NSW 2560, Australia; K.agho@westernsydney.edu.au; 4School of Health Sciences, Western Sydney University, Penrith, NSW 2751, Australia; 5Fall, Balance and Injury Research Centre, Neuroscience Research Australia, Randwick, NSW 2031, Australia; L.harvey@neura.edu.au; 6School of Population Health, University of New South Wales, Kensington, NSW 2033, Australia

**Keywords:** underreported residential fire incidents, linked data, health economics, epidemiology

## Abstract

The rate of fires, and particularly residential fires, is a serious concern in industrialized countries. However, there is considerable uncertainty regarding the reported numbers of residential fire incidents as official figures are based on fires reported to fire response agencies only. This population-based study aims to quantify the total number of residential fire incidents regardless of reporting status. The cohort comprised linked person-level data from Fire and Rescue New South Wales (FRNSW) and health system and death records. It included all persons residing at a residential address in New South Wales, Australia, that experienced a fire between 1 January 2005 and 31 December 2014. The capture-recapture method was used to estimate the underreporting number of residential fire-related incidents. Over the study period, 43,707 residential fire incidents were reported to FRNSW, and there were 2795 residential fire-related health service utilizations, of which 2380 were not reported. Using the capture-recapture method, the total number of residential fire incidents was estimated at 267,815 to 319,719, which is more than six times the official records. This study found that 15% of residential fire incidents that were identified in health administrative dataset were reported. The residential fire incidents that were not reported occurred mainly in socio-economically disadvantaged areas among males and adults.

## 1. Introduction

Fires are potentially devastating events globally, resulting in an estimated 265,000 deaths each year [1]. Although fire-related injuries and fatalities are considerably more common in low-income countries, the issue is highly relevant to high-income countries [2,3,4]. Fire-related burns are ranked as the fourth most common cause of unintentional trauma globally [2], and in most industrialized countries, a great proportion of fire-related deaths and injuries are related to residential fires [5]. In the US, there were 363,000 residential fires in 2018 which caused 2720 deaths and 11,200 injuries, which accounted for 93% and 88% of fire deaths and injuries, respectively [6].

In Australia, similar to other countries, residential fire poses a significant burden. According to the most recent available data between 2019 and 2020, there were 17,915 accidental residential fire incidents, of which 6591 occurred in New South Wales [7]. NSW is the most populous state in Australia, with a total population of over eight million residents [8], where residential fires lead to approximately 20 fatalities and 550 injuries each year [9].

Globally, researchers have used different approaches and data sources to determine residential fire-related incidents and related deaths and injuries. The most common data source that has been used in residential fire-related national and international studies is fire department data [10,11,12,13,14,15,16,17].

In some studies, medical examiner records and cause of death registry data were used to identify residential fire-related incidents [18,19]. Alternatively, police report and national census data were used to determine the incidents that were caused by residential fires [20,21,22]. Some studies used questionnaires and surveys to estimate the number of residential fire-related incidents, including [17,23,24] and some used two or further databases to provide a better picture of the residential fire-related incidents and their outcome. In Australia, Xiong et al. [25] used residential fire-related deaths from the coronial data as well as interviews from the survivors of residential fires, and in a matched case control Swedish study, Jonsson and Jaldell [26] used census data and the national database on fatal fires to determine the number of residential fire-related deaths. DiGuiseppi et al. [27], in a population-based study, identified fire-related injuries by screening routinely collected data from a variety of sources, including local emergency departments and hospitals, ambulance and helicopter services, the fire department and the local coroner.

Some studies used linked data to extract additional information regarding the residential fire incidents and the individual’s characteristics. Istre et al. [28] conducted a study in the US to describe the epidemiology of residential fire-related deaths and injuries among children and its associated risk factors. They used fire department records linked with ambulance transports, hospital admissions and medical examiner records. In Sweden, Jonsson et al. [29] used linked data including the database on fatal fires, the database on forensic examinations, and the cause of death register. They described the epidemiology of fatal residential fires and described the common features in terms of both the characteristics of the individuals involved and the characteristics of the event itself.

In Australia, residential fire-related studies have mainly used fire department data from Fire & Rescue New South Wales [9], coronial records from the National Coronial Information System (NCIS) database [30,31], Australian Bureau of Statistics (ABS) data [32,33,34] and primary health networks (PHNS) data [35,36]. In addition, population health survey data have been used for residential fire-related studies [24].

According to recent research studies, the reported number of residential fires, fire-related injuries and deaths significantly underestimated the true number [23,24,37]. In Greene and Andres [23], US surveys conducted in 1974, 1984, and 2004–2005 demonstrated that the majority of residential fires were not reported to or attended by the fire department. In addition, Chubb [37] showed that many individuals who access medical treatment for fire-related injuries in the US or New Zealand do not have an associated fire incident report. A NSW population survey-based study by Tannous and Agho [24] reported that around one third of respondents who experienced a residential fire did not contact fire services. The survey contained self-reported information about respondents’ socio-demographic characteristics, including household size, the respondent’s age, gender, level of education, employment status, income, ethnicity, socioeconomic status and smoker status. Additionally, respondents were asked about fire incidents including “Have you ever experienced an unintentional or accidental fire in your home?” and “Was the fire brigade called to put out the fire?”

Tannous and Agho [24] identified factors associated with unwillingness to contact fire services including females, high household income (AUD 80,000 plus), having both a battery and hardwired smoke alarm installed at home, and speaking a language other than English at home. Furthermore, the study detailed that the small size of fires and early warning of smoke alarms can result in individuals extinguishing fires and may not report the incidents to the local fire brigade. In that study, the authors noted a number of limitations. These include the use of data from the NSW Population Health Survey, and hence responses to questions on whether they had ever experienced a residential fire were self-reported. The survey was cross-sectional and relied on recollections of past fire events and some may have occurred years prior. Tannous and Agho [24] recommended the use of administrative data linked across Fire and Rescue NSW (FRNSW), the state’s urban fire brigade, and health services, as a number of respondents had stated that they had experienced a major residential fire incident that resulted in significant injuries, but they did not contact the fire response agency.

Australian Incident Reporting system (AIRS) data include all the information and statistics regarding fire incidents collected by Fire and Rescue NSW (FRNSW). This includes fire incidents that were attended by FRNSW and therefore, FRNSW does not have statistics on unreported residential fire incidents and related deaths and injuries. It has been noted that even though AIRS also collects data from different organizations including emergency services, hospitals and insurance companies, their knowledge regarding unreported residential fires is likely to be incomplete [9,33].

To address the limitation in the literature, FRNSW AIRS data were linked with health administrative datasets in NSW for the 10-year period of 2005–2014. This study is a follow-up to Tannous and Agho’s study [24] and the main aim is to calculate the number of residential fires in NSW across different linked datasets including FRNSW AIRS data, ambulance data sources, emergency department, hospital admissions, out-patient burn clinics and mortality data sources. In addition, it will provide an estimate of the unreported residential fire incidents and their scale across different datasets that is applicable to other states and countries.

The number of officially reported residential fire incidents is used for the planning and funding of government response services and organizations. In addition, it is also used for community engagement and education programs. Thereby, obtaining a true number of residential fire incidents by severity is vital for this service delivery planning. These results will assist in better allocating resources to respond to residential fire incidents. In addition, it will provide direction for prevention strategies, policies and regulations for the built environment and individual behavior attributes.

## 2. Materials and Methods

### 2.1. Study Population and Data Sources

The study cohort included all persons residing in NSW who experienced a residential fire in the period of 1 January 2005 to 31 December 2014. This included any person whose residential fire-related injury or death was identified through linked administrative health datasets, irrespective of whether they had an associated residential fire incident record. The protocol for this linkage study has been detailed elsewhere [38].

Nine data sources were linked to identify the individual’s presentation in either of the emergency services or health providers. The emergency services data sources were: the FRNSW Australian Incident Reporting System (FRNSW AIRS) and the Computer Aided Dispatch (CAD) system; the ambulance paper-based Patient Health Care Record (PHCR) and the electronic medical record (eMR). The hospital data included NSW Emergency Department Data Collection (EDDC), Admitted Patient Data Collection (APDC) and for outpatient burns patients, the NSW State-wide Burn Injury Service (SBIS). For mortality data, the Registry of Births, Deaths and Marriages (RBDM) and the Australian Bureau of Statistics (ABS) Cause of Death Unit Record File (COD-URF) were used to provide a complete study cohort (Table 1).

Linkage was undertaken by the Centre for Health Record Linkage (CHeReL) using probabilistic matching [39]. Five of the datasets (NSW Ambulance (CAD, EMR, PHCR), APDC, EDDC, NSW RBDM, NSW COD-URF) are held in the CHeReL Master Linkage Key (MLK), a system of continuously updated links between core health-related datasets in NSW [39]. For this study, records for the various hospital admissions, emergency department, deaths registry and ambulance data sources were extracted from the MLK, and the MLK extract was linked to the FRNSW AIRS and SBIS records. Once the linkages were finalized, the CHeReL created a project person number (PPN), a unique person identifier for each person in the linkage, and assigned this PPN to the records.

### 2.2. Case Selection and Creation of Study Variables

The number of the residential fire incidents were identified using two different case selection criteria.

#### 2.2.1. Selection Criteria 1: Identifying “Residential Fire-Related Incidents” in Each Dataset

From the FRNSW AIRS, which contains the response agency incident data, the ‘type of property’ variable was used to identify records of residential fire incidents.

The NSW ambulance data collections contain operational information from the CAD system, and also data documented by clinicians in the PHCR and clinical and treatment information on the patients in the eMR. The ‘case given as’ and ‘main condition’ variables were used to identify “Building Fire” and “House Fire” records.

The NSW EDDC uses three different types of clinical coding, with presenting diagnosis recorded using either the Australian version of the International Classification of Disease and Related Problems (ICD), ninth revision, Clinical Modification (ICD-9-CM), or the Australian Modification of the 10th revision, (ICD-10-AM), or the Systematized Nomenclature of Medicine, Clinical Terms (SNOMED-CT). The ‘principal diagnosis code’ variables were used to identify residential fire-related records, noting that the use of principal diagnosis with ICD codes identifies the main reason of ED admission of burn or smoke inhalation and does not provide details about the type of fire. The different SNOMED codes used to identify residential fire-related records in this study are detailed in Table 2.

The APDC data that contain all the records of hospital admissions with ‘principal diagnosis’ and ‘external cause’ coded using ICD-10-AM and ICD-10-AM codes denoting “exposure to uncontrolled building fire’” and “exposure to controlled building fire” were used to identify residential fires (Table 2). From the SBIS data, which contain admission records and case details for all patients admitted to the three designated burn units in NSW (two adult, one pediatric) and capture both hospitalizations and outpatient clinical visits, the free text description of the ‘mechanism of injury’ variable was used to identify residential fire-related records.

The mortality dataset, including the RBDM and the COD-URF, contains records of all deaths of NSW residents, whether certified by a registered medical practitioner or by the state coroner, and underlying causes of death coded using ICD-10-CM, respectively. ‘Diagnosis code’ and ‘place of occurrence’ variables were used to identify residential fire-related deaths (Figure 1).

The records in each dataset that were identified as residential fire-related incidents were flagged. Hospitalizations and health service utilization records with the same diagnosis codes for the same individuals within a few days were considered as one residential fire incident. Residential fire-related records with missing ‘Date of incident’ or ‘PPN’ were removed, as there was no way to determine whether they belonged to our study timeframe or had already been counted in other datasets.

Residential fire-related incidents in FRNSW AIRS were identified using the ‘Incident ID’ variable that represented a residential fire incident that could involve one or more individuals each with a unique PPN sharing the same incident ID. Residential fire-related incidents in other datasets were identified based on the individuals using health services as a result of residential fire. However, since other datasets did not have the Incident ID variable, to decide whether the injuries of multiple individuals in a single day belong to one residential fire incident, the location of each incident was estimated using variables such as ‘postcode’ or statistical area level 2 ‘SA2′ of the individuals 

Socio-Economic Indexes for Areas (SEIFA), an index developed by the Australian Bureau of Statistics to rank localities in Australia according to their relative socio-economic advantage and disadvantage, was used and merged with other datasets based on postcode or statistical area level 2 (SA2). SEIFA uses a broad definition of relative socioeconomic disadvantage in terms of people’s access to material and social resources and their ability to participate in society, and represents an average of all people living in an area rather than the individual situation of each person. The Index of Relative Socioeconomic Advantage and Disadvantage (IRSAD) deciles were adopted to rank localities where fires occurred. Deciles divide a distribution into 10 equal groups, with a number of 1 referring to the lowest scoring 10% of areas, while 10 refers to the highest 10% of areas. SEIFA was adopted in this study to determine whether underreported residential fire incidents occurred in the socioeconomically disadvantaged areas.

#### 2.2.2. Selection Criteria 2: Identifying Unreported Residential Fire Incidents by Merging Each Dataset with FRNSW AIRS Dataset

Individuals involved in residential fire incidents reported to FRNSW and who used any health services or died were identified by merging the FRNSW AIRS dataset with the other datasets using ‘date’ and ‘PPN’ within the two-week time period from the date of the fire incident to the date of presentation to or using health services or death registration. Burn experts suggest the inclusion of a two-week lag-period between the date of the fire event and health service use as some individuals might prefer not to use ambulance and health services immediately following the incident and delay seeking help. Patients with moderate injuries usually use health services within 3–7 days; however, an additional week is recommended to account for individuals with less severe injuries or those who tried home remedies before seeking medical care. If individuals require health services after two weeks from the incident date, their injuries are considered unrelated to that incident. 

These records that were identified in the AIRS dataset were flagged as “Reported incidents” and were merged with the records that were flagged as “Residential fire-related incidents”. “Unreported residential fire-related incidents” were then identified (Figure 1).

### 2.3. Statistical Analysis

#### 2.3.1. Estimation of Incidence

To determine the number of residential fire incidents that used health services, we used the data derived from selection criteria 1 and selection criteria 2, as shown in Figure 2 where;

M: All the residential fire-related incidents reported to FRNSW that were extracted from FRNSW AIRS data

n: All the residential fire-related incidents extracted from other datasets (excluding FRNSW AIRS data)

m: Records that were common in both M and n

The total number of the health service utilization records as a result of residential fire-related incident = (M ꓴ n) − m, which is the union of two sets of M and n subtracted by the records that are common in both M and n,

The total number of unreported health service utilization records as a result of a residential fire-related incident = n − m.

#### 2.3.2. Capture-Recapture Method (CR)

The capture-recapture method is a technique that can be used to estimate the total population by counting people captured by each system and the extent of data overlap. It was first introduced by Lincoln Petersen in 1896, and since 1949 this model has become a standard tool to characterize prevalence in human population [40].

In order to use the capture-recapture method for population estimation, based on different data sources, the data sources should represent approximately the same population at approximately the same time period. In addition, to obtain reliable results, certain assumptions must be met. This includes the assumption that data sources should be independent, which means the probability of being in a data source should not increase or decrease the probability of being in the others. The second assumption is that the probability of association within each source or catchability is equal for all individuals and factors such as age or gender does not affect their probability of being captured in those data sources. Finally, the population must be closed, which means it is assumed that it does not increase by birth or decrease by deaths. According to Petersen’s estimator, the total number of a population is determined by the product of the two data sources divided by the number of cases that were common in both data sources:$$\mathrm{Petersen}\;\mathrm{estimator}\;N=\frac{\mathrm{Mn}}{m}$$

However, Petersen’s estimator is subjected to bias if the common cases are small or zero. In 1951, therefore, Chapman modified the Petersen’s estimator [41].
Chapman’s estimator N = [
(M+1)(n+1)
(m+1)
]−1

As shown in Figure 2;

M: FRNSW AIRS data = all the reported fire incidents to FRNSW

n: all the fire incidents that were derived from the health administrative data

m: all the residential fires that were captured in both health administrative data being identified as residential fire-related records and in FRNSW AIRS dataset (reported to FRNSW)

The following formulas were used to calculate the standard error of the estimates (SE), and the confidence interval of the estimates (95% CI).
$$\mathrm{SE}(N)=\sqrt{\frac{\left(N-M\right)\left(N-n\right)}{m}}$$
95%CI = N ± 1.96SE

The statistical software used for this study were SAS version 9.2 [42] and R-3.6.0 [43].

## 3. Results

For the period of 2005 to 2014, 43,707 residential fire incidents were reported to FRNSW, involving 43,433 individuals. Using the residential fire-related identification method for each dataset, 2380 additional residential fire-related incidents were identified. This includes 1200 residential fire incidents in the NSW Ambulance dataset, 11 residential fire incidents in the emergency department, 1551 residential fire incidents in the hospital and 404 residential fire incidents in the burns outpatient clinic dataset. In addition, 142 residential fire incidents were identified in the mortality dataset. Only 415 incidents that were identified in health administrative datasets using residential fire identifiable variables were also captured in FRNSW AIRS, equating to only 15% (n = 415 out of n = 2795) of the residential fire-related health service utilization records that were reported to the fire agencies (Figure 1).

Using the capture-recapture model, the total number of the residential fire incidents was estimated at between 267,815 to 319,719 incidents over the study period. This was determined as follows;

M: Residential fire incidents in FRNSW AIRS data = 43,707;

n: all the residential fire incidents that were identified in other datasets from their variables =2795;

m: all the residential fire-related incidents that were common in M and n = 415;
$$\mathrm{Chapman}’s\;\mathrm{estimator}\;N=[\;\frac{\left(M+1\right)\left(n+1\right)}{\left(m+1\right)}\;]-1=[\;\frac{\left(43,707+1\right)\left(2795+1\right)}{\left(415+1\right)}\;]-1=\;293,767$$
$$\mathrm{SE}(N)=\sqrt{\frac{\left(N-M\right)\left(N-n\right)}{m}}=\mathrm{SE}(N)=\sqrt{\frac{\left(293,767-43,707\right)\left(293,767-2795\right)}{415}}=13,241$$
95%CI = N ± 1.96SE = 293,767 + 1.96(13,241) = 319,719
=293,767−1.96(13,241) = 267,815

This translates to an average of 26,782 (n = 267,815 in 10 years) to 31,972 (n = 319,719 in 10 years) residential fire incidents per year and compares to the official records of around 4371 (n = 43,707 in 10 years) residential fire incidents per year. It is noted that the official number is at 15% (n = 43,707 out of N = 293,767) of the determined figure, with 85% of the residential fire incidents going unreported.

Table 3 illustrates demographic information including age, gender, country of birth and socio-economic status for those who did not have an associated residential fire report and used health services. The unreported residential fire incidents mainly occurred among adults (33.5%) who were unmarried (24.2%) and Australian born (42.2%). The rate of unreported residential fires was higher among men (35.2%) and those who lived in socio-economically disadvantaged areas.

Figure 3 shows the number of fire incidents using FRNSW AIRS data per year and compares them with the figures provided in the Australian Productivity Commission report [44]. As shown, FRNSW AIRS data cover the residential fire incidents in the major cities, including two thirds of the incidents in NSW. However, in this study, we estimated that the number of residential fire incidents is more than six times the total number of residential fire incidents in NSW within the study period.

## 4. Discussion

This study demonstrated that a high portion of residential fire incidents that required health services were not reported and were not reflected in official records. This finding is consistent with previous studies in NSW, a self-reported study by Tannous and Agho [24] and international studies regarding underreported records and using health services without having an associated fire incident report [24,37,45,46].

This is a whole population-based study that has provided data on fire incidents by linking datasets from diverse systems that had never previously been linked including: FRNSW AIRS, NSW ambulance (CAD, PHCR, eMR) data sources, emergency department (NSW EDDC) data, hospital (NSW APDC) data, outpatient burns clinics (SBIS) data and mortality data, including RBDM and COD-URF.

Using linked data, we created a comprehensive database on residential fires in NSW to provide a better understanding of the total number of residential fire incidents and underreported residential fires. In this study, integrating the FRNSW AIRS data with the administrative health datasets showed that a great portion of residential fire-related health services use are not being able to capture by just referring to the fire brigade’s report and their data resources.

The implications of underreporting residential fire incidents are most significant as these statistics provide the evidence to direct policy makers in allocating proper resources, such as firefighting and prevention plans, with an absolute understanding of the residential fire risk and its ongoing impact on people’s lives at the individual and community level.

### Strengths and Limitation of the Study

The strength of this study is that it used data linkage, which is the best available method to identify and quantify the total number of residential fire incidents. As discussed and shown, referring to fire agency services to report residential fire incidents would simply mean neglecting unreported records that affect the true number of residential fire incidents, and hence residential fire-related deaths and injuries as well as health service utilization cost. This emphasizes the importance of data linkage to determine the health impact and health service utilization cost.

In this study, two-week timeframe sensitivity analysis from the day of incident was used, through which all the residential fire-related incidents that were reported to FRNSW and used health services within the next two weeks were identified. The number of records using health services for the reported incidents increased as we varied the time frame from the day of incident to a two-week time period to avoid missing any reported fire incident records that could affect the estimation of residential fire incidents. (Figure 1).

This study has several limitations, however, which are inherent in data linkage studies that rely on administrative data. This is the first time that CHeReL has linked health administrative data with a property-based dataset of FRNSW. The core business of FRNSW is to focus on attending incidents and managing fire at a place. Their administrative data contain detailed information on the structure of the property that was attended. The collection of persons’ data that were involved in an incident regarding the structure is minimal, as it is not the primary concern. Therefore, estimating the true number of individuals involved in residential fire incidents is challenging. Additionally, the data that are collected by FRNSW with regard to the individuals is quite limited and does not include their identifiable details of full legal names and date of birth required for probabilistic matching. As such, the linking of individuals to a residential fire incident and other administrative datasets was based on address and date of incident (within 14 days). The process of probabilistic linking of fire brigade incident data with health service utilization and death records has not been done previously, and this is a major strength of this study, with the recognition of the limitations of linking this response to agency property data.

FRNSW AIRS data include fires in the major cities in NSW and do not cover fires in the remote and rural areas that were attended only by NSW Rural Fire Services (NSWRFS). This can impact the total number of fire incidents and therefore the unreported residential fire incidents. This also could mean that some of those who used health services might have reported their residential fire incident directly to NSWRFS (FRNSW is the Triple Zero (000) call-taking agency for fires in NSW). In addition, the quality of administrative health data and their coding is another limitation of this study.

For this study, primary care records were not linked, as the reason for visiting general practitioners (GPs) is not recorded. In addition, GP and medical center data are held by the federal health department, meaning they are not easily accessible for researchers due to time and cost. Therefore, those who experienced a residential fire incident and accessed only general practitioners, medical centers or pharmacies to treat their injuries were not included in this study. However, burns managed by a GP or pharmacist are likely to be minor in nature and there are well documented guidelines for referral to a specialist outpatient burn clinic for more severe injuries.

Additional linkage of NSW Rural Fire Services data as well as general practice (GP) usage from the Medicare Benefit Schedule (MBS) and medication use from the Pharmaceutical Benefits Scheme (PBS) to our existing data could be feasible solutions to several of the limitations of our study. However, the additional information obtained and the lack of specificity for the reason for GP management available from the MBS would be difficult to attribute to the fire incident and associated injuries.

Given the limitations of using stand-alone administrative data highlighted by this study, future research using multiple linked datasets would be beneficial to fully identify and accurately quantify individuals’ health care use and long-term outcomes following residential fire. This would enable research into the impact on the individuals beyond immediate clinical care and on the wider economy.

## 5. Conclusions

This study found that only 15% of residential fire incidents that required health utilization services were reported. The residential fire incidents that were not reported occurred mainly in socioeconomically disadvantaged areas among males and adults. The better identification of disadvantaged populations helps with the allocation of firefighters and fire prevention resources.

Our findings are consistent with the existing previous work on individuals’ unwillingness to report residential fires in Australia and other countries, emphasizing that government education and prevention policies are based on significantly lower numbers that those that actually affect residential fire safety prevention.

This highlights the importance of data linkage for accurate communication to policy makers and the public on the prevalence and impact of residential fires. The absence of good-quality and consistent data makes the determination of residential fire impacts, as well as the identification of trends and prevention strategies, impossible for authorities.

## Figures and Tables

**Figure 1 ijerph-18-06921-f001:**
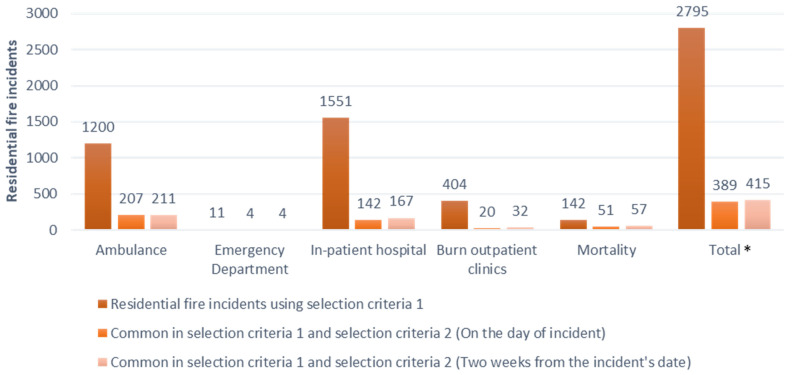
Number of residential fire-related records in all the datasets except FRNSW AIRS (Criteria 1) and residential fire-related incidents identified in both FRNSW AIRS data and other datasets after being merged with FRNSW AIRS (Criteria 2); residential fire incidents in FRNSW AIRS = 43,707 incidents, NSW, 2005–2014. * Note: Some records were identified in more than one dataset and were repeated in different datasets, and therefore the total value is different from the sum of the fire incidents in each dataset. Number of unreported residential fire incidents = 2795 − 415 = 2380.

**Figure 2 ijerph-18-06921-f002:**
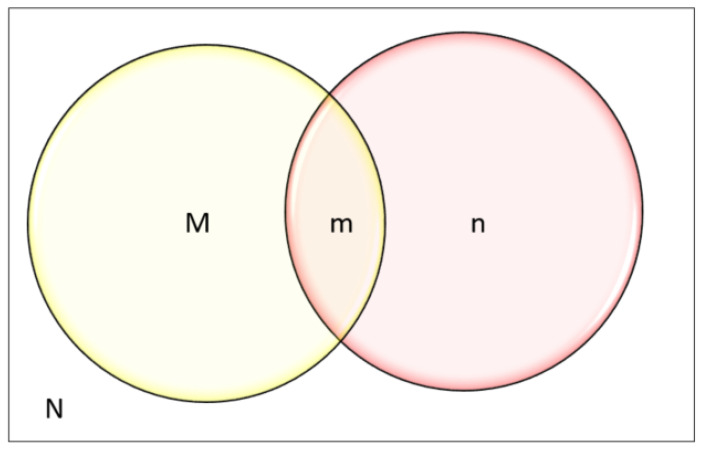
Venn diagram for Chapman’s estimator to determine the total population N, where M: FRNSW AIRS data = all the individuals who had a residential fire incident and reported, n: all the individuals who used health services as a result of residential fire.

**Figure 3 ijerph-18-06921-f003:**
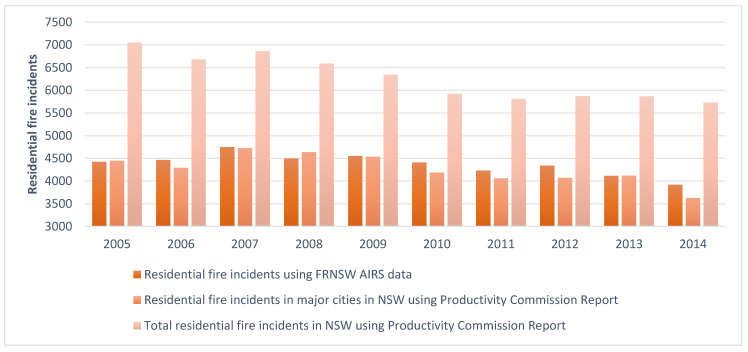
Number of residential fire incidents per year using FRNSW AIRS data vs. the Australian Productivity Commission Report in NSW and Major Cities of NSW.

**Table 1 ijerph-18-06921-t001:** Method of identification of residential fire-related records in each dataset, NSW, 2005–2014.

List of Agencies	Dataset	Method of Identification	Variables
Fire and Rescue NSW	FRNSW AIRS	Type of property variable to be “residential buildings”	‘type of property’ to be residential
Ambulance	CAD EMR PHCR	Key words such as “building fire”, “residential fire” and “House fire” as the reason for using ambulance in each dataset	‘main condition’ and ‘case given as’ variables
Emergency Department	NSW EDDC	“house fire”, “residential fire” and “conflagration in private dwellings”	‘principal diagnosis code’ ICD-9-CM, ICD-10-AM and SNOMED CT
In-patient hospital	NSW APDC	Exposure to uncontrolled/controlled fire in building or structure	‘external cause’ ICD-10-AM codes “X00” and “X02” in external cause code variables
Burns outpatient clinics	SBIS	“house fire” and “residential fire”	Looking up words such as “house fire” in ‘mechanism of injury’ free text variable
Mortality	RBDM COD_URF	Exposure to uncontrolled/controlled fire in building or structure	ICD-10-CM codes” X00” and “X02” in ‘external cause’ variables and place of occurrence to be “home”

**Table 2 ijerph-18-06921-t002:** List of codes in SNOMED CT and ICD-10-AM/ICD-10-CM.

Codes	Definition
SNOMED CT	
242430009	House fire
269776001	House fire—accident
309728007	Accidental caused by fire and flames—fire in private dwelling
157906004	Fire in private dwelling
269777005	Fire in building
217177006	Conflagration in private dwelling (and house fire)
257195008	Fire—domestic object (including electric fire, gas fire, solid fuel fire)
257199002	Domestic heating appliances—house hold accessory
783264007	Fire door—domestic structure-doorways and door
257203302	Domestic heating appliances—house hold accessory—gas
30002008	Domestic heating appliances—house hold accessory—fireplace
257200004	Domestic heating appliances—house hold accessory—enclosed gas fire
257204008	Domestic heating appliances—house hold accessory—open domestic fire—open charcoal fire
257205007	Domestic heating appliances—house hold accessory—open domestic fire— open coal fire
257206005	Domestic heating appliances—house hold accessory—open domestic fire—open coal, charcoal, wood fire
257207001	Domestic heating appliances—house hold accessory—open domestic fire—open gas fire
257208006	Domestic heating appliances—house hold accessory—open domestic fire—open wood fire
257210008	Domestic heating appliances—house hold accessory—solid fuel fires (enclosed solid fuel fire, open domestic fire)
257201000	Domestic heating appliances—house hold accessory—enclosed solid fuel fire
ICD-10-AM/ICD-10-CM	
X00	Exposure to uncontrolled fire in building or structure
X02	Exposure to controlled fire in building or structure

**Table 3 ijerph-18-06921-t003:** Demographics of the individuals who used health services and did not report their residential fire incident to FRNSW, NSW, 2005–2014.

Demographics	Number	(%)
Age Group		
Infant (<1 year)	0	0.0
Toddler (1–4 years)	74	3.1
Child (5–14 years)	86	3.6
Youth (15–24 years)	192	8.1
Adult (25–64 years)	797	33.5
Elderly (65+ years)	254	10.7
Unknown	977	41.0
Total	2380	100.0
Marital Status		
Never Married	575	24.2
Married	458	19.2
Divorced/Separated/Widowed	242	10.2
Unknown	1105	46.4
Total	2380	100.0
Gender		
Male	838	35.2
Female	553	23.2
Unknown	989	41.6
Total	2380	100.0
Country of birth		
Australian	1004	42.2
Non-Australian	326	13.7
Unknown	1050	44.1
Total	2380	100.0
SEIFA		
1 (The most disadvantaged)	203	8.5
2	145	6.1
3	150	6.3
4	118	5.0
5	132	5.5
6	130	5.5
7	119	5.0
8	93	3.9
9	104	4.4
10 (The most advantaged)	78	3.3
Unknown	1108	46.5
Total	2380	100.0

## Data Availability

The data presented in this study are not publically available due to sensitive health and personal data and medical confidentiality.

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
