# Peer review of "Estimating the Total Number of Residential Fire-Related Incidents and Underreported Residential Fire Incidents in New South Wales, Australia by Using Linked Administrative Data"

_ijerph, 2021, doi:10.3390/ijerph18136921_

Round 1

Reviewer 1 Report

This study found that only 15% of residential fire incidents that required health utilization services were reported. The residential fire incidents that were not reported occurred mainly in socioeconomically disadvantaged areas among the males and adults. The better identification of disadvantaged populations helps with allocation of firefighters and fire prevention resources. The finding highlights the importance of data linkage for accurate communication to policy makers and the public on the prevalence and impact of residential fires. This paper is very interesting and may be useful to the administrative firefighting policy.

Author Response

Thank you kindly for your time and feedback - No revisions requested.

Reviewer 2 Report

This paper estimating the total number of residential fire-related incidents and underreported residential-fire incidents in New South Wales, Australia by using linked administrative data. The topic is interesting. The author innovatively uses capture-recapture method to estimate the underreporting number of residential fire-related incidents. Detailed comments are as follows.

  • The first paragraph of the text, line 39-41, if the authors can use some data to quantify the occurrence of residential fires will be more persuasive.
  • In this paper, many tables are used to explain the data, but the tables are lack intuition. If the authors can use the combination of the charts (for example, use a histogram) and tables to discuss will be welcome.
  • Line 236-238, it mentioned “Inclusion of a two-week lag-period between date of fire event and health service use (ED and hospital) was based on recommendations from expert Burn clinicians, to account for delayed presentation for care”. Can the “two-week lag-period” be explained more clearly. Why two weeks? Can you elaborate on what the experts suggest? What are the reasons for the delay in providing care?
  • Section 4.1, this section discusses the strengths and limitation of this study, can the author put forward some feasible solutions according to the limitations?
  • According to the existing problems in this study, can the author give the future research prospects in the end?

Author Response

Thank you Kindly for your time and feedback.

Reviewer 3 Report

The paper concerns the problem of the estimation of the total number of residential fires. Regarding the data presented in the literature review the estimations based on the fire departments datasets are underestimated. Therefore the authors examine the problem against the various data sources.

I have not major concerns regarding the paper. The experiment was properly designed, conducted, and described. The literature review related to the problem in question was also comprehensive.

My only concern is related to the better explanation in the paper of how the number of people affected by the fire incidents is transformed into the number of incidents. This was not clearly described in the paper. Also, the discussion related to uncertainty to this estimation (number of people to the number of the incidents) should be discussed.

Author Response

Thank you kindly for your time and feedback.
